# The Importance of Research on the Origin of SARS-CoV-2

**DOI:** 10.3390/v12111203

**Published:** 2020-10-22

**Authors:** Kenneth Lundstrom, Murat Seyran, Damiano Pizzol, Parise Adadi, Tarek Mohamed Abd El-Aziz, Sk. Sarif Hassan, Antonio Soares, Ramesh Kandimalla, Murtaza M. Tambuwala, Alaa A. A. Aljabali, Gajendra Kumar Azad, Pabitra Pal Choudhury, Vladimir N. Uversky, Samendra P. Sherchan, Bruce D. Uhal, Nima Rezaei, Adam M. Brufsky

**Affiliations:** 1PanTherapeutics, CH1095 Lutry, Switzerland; 2Doctoral Studies in Natural Sciences and Technology SPL44, University of Vienna, 1010 Vienna, Austria; muratseyran@gmail.com; 3Network of Immunity in Infection, Malignancy and Autoimmunity (NIIMA), Universal Scientific Education and Research Network (USERN), Vienna, 1010 Vienna, Austria; 4Department of Global Health, Italian Agency for Development Cooperation—Khartoum, Al Amarat 111111, Sudan; damianopizzol8@gmail.com; 5Department of Food Science, University of Otago, Dunedin 9054, New Zealand; pariseadadi@gmail.com; 6Department of Cellular and Integrative Physiology, University of Texas Health Science Center at San Antonio, San Antonio, TX 78229-3900, USA; mohamedt1@uthscsa.edu (T.M.A.E.-A.); soaresa@uthscsa.edu (A.S.); 7Zoology Department, Faculty of Science, Minia University, El-Minia 61519, Egypt; 8Department of Mathematics, Pingla Thana Mahavidyalaya, Maligram, Paschim Medinipur, West Bengal 721140, India; sarimif@gmail.com; 9Department of Applied Biology, CSIR-Indian Institute of Chemical Technology, Tarnaka, Hyderabad 500007, India; ramesh.kandimalla@gmail.com; 10Department of Biochemistry, Kakatiya Medical College/MGM-Hospital, Hyderabad 500007, India; 11School of Pharmacy and Pharmaceutical Science, Ulster University, Coleraine BT52 1SA, Northern Ireland, UK; m.tambuwala@ulster.ac.uk; 12Department of Pharmaceutical Sciences, Faculty of Pharmacy, Yarmouk University, Irbid 21163, Jordan; alaaj@yu.edu.jo; 13Department of Zoology, Patna University, Patna, Bihar 800005, India; gkazad@patnauniversity.ac.in; 14Applied Statistics Unit, Indian Statistical Institute, Kolkata, West Bengal 700108, India; pabitrapalchoudhury@gmail.com; 15Department of Molecular Medicine, Morsani College of Medicine, University of South Florida, Tampa, FL 33612, USA; vuversky@usf.edu; 16Department of Environmental Health Sciences, Tulane University, New Orleans, LA 70112, USA; sshercha@tulane.edu; 17Department of Physiology, Michigan State University, East Lansing, MI 48824, USA; bduhal@gmail.com; 18Center for Immunodeficiencies, Pediatrics Center of Excellence, Children’s Medical Center, Tehran University of Medical Sciences, Tehran 1419733151, Iran; rezaei_nima@yahoo.com; 19Network of Immunity in Infection, Malignancy and Autoimmunity (NIIMA), Universal Scientific Education and Research Network (USERN), Tehran 1419733151, Iran; 20UPMC Hillman Cancer Center, Department of Medicine, Division of Hematology/Oncology, University of Pittsburgh School of Medicine, Pittsburgh, PA 15213, USA; brufskyam@upmc.edu

**Keywords:** coronavirus, COVID-19 pandemic, origin of SARS-CoV-2, receptor binding domain, genome homology, natural selection, zoonotic transfer

## Abstract

The origin of the severe acute respiratory syndrome-coronavirus 2 (SARS-CoV-2) virus causing the COVID-19 pandemic has not yet been fully determined. Despite the consensus about the SARS-CoV-2 origin from bat CoV RaTG13, discrepancy to host tropism to other human Coronaviruses exist. SARS-CoV-2 also possesses some differences in its S protein receptor-binding domain, glycan-binding N-terminal domain and the surface of the sialic acid-binding domain. Despite similarities based on cryo-EM and biochemical studies, the SARS-CoV-2 shows higher stability and binding affinity to the ACE2 receptor. The SARS-CoV-2 does not appear to present a mutational “hot spot” as only the D614G mutation has been identified from clinical isolates. As laboratory manipulation is highly unlikely for the origin of SARS-CoV-2, the current possibilities comprise either natural selection in animal host before zoonotic transfer or natural selection in humans following zoonotic transfer. In the former case, despite SARS-CoV-2 and bat RaTG13 showing 96% identity some pangolin Coronaviruses exhibit very high similarity to particularly the receptor-binding domain of SARS-CoV-2. In the latter case, it can be hypothesized that the SARS-CoV-2 genome has adapted during human-to-human transmission and based on available data, the isolated SARS-CoV-2 genomes derive from a common origin. Before the origin of SARS-CoV-2 can be confirmed additional research is required

The COVID-19 pandemic has seriously touched the whole world with over 40 million infections and claiming more than 1 million lives as of today (21 October 2020), also causing social and economic havoc globally. There is currently a tremendous amount of both competitive and collaborative efforts in search of novel antiviral drugs and vaccines reaching almost desperate proportions. Among all this, the question of the origin of the “culprit”, the severe acute respiratory syndrome-coronavirus 2 (SARS-CoV-2), is being addressed. Numerous politically motivated conspiracy theories have surfaced, and hypotheses have arisen, including the unintentional or intentional escape/release of the virus from a high-security laboratory facility in Wuhan, China. For instance, it was reported [1] (now withdrawn) that SARS-CoV-2 contained four inserts in its spike (S) glycoprotein, critical for virus entry, which were either identical or similar to motifs found in the Env and Gag proteins of HIV-1. It was speculated that fragments from the HIV-1 genome had been intentionally introduced into the SARS-CoV-2 genome. A thorough bioinformatics analysis and sequence examination of SARS-CoV-2, other Coronaviruses, and HIV-1 from the GenBank database demonstrate that there is currently no compelling evidence that HIV-1 specific inserts in SARS-CoV-2 exist [2]. Additional analysis initially suggests that the probability is low that SARS-CoV-2 is a laboratory construct or intentionally engineered [3].

The self-assembled COVID consortium, consisting of international experts in bioinformatics, structural biology, molecular biology, immunology, and virology, has just published a Letter in the Journal of Medical Virology [4] in response to publications on the natural origin of SARS-CoV-2. It stated that despite the consensus of SARS-CoV-2 originating from bat CoV RaTG13, SARS-CoV-2 had demonstrated significant discrepancies to other human Coronaviruses related to host tropism. Moreover, bat and rodent Coronaviruses have seen some specific changes in the S protein receptor-binding domain (RBD) and the glycan-binding N-terminal domain (NTD) in host tropism [5,6]. However, SARS-CoV-2 sequences do not contain these changes, indicating a very recent origin of RBD and NTD subdomains. For instance, the hidden glycan-binding domains located in cavities in the S protein NTD domain, limiting their access to antibodies and immune cells, are not present in SARS-CoV-2 [6]. The surface of the sialic acid-binding domain of the SARS-CoV-2 S protein is flat and non-sunken, unlike other Coronaviruses, influenza viruses, rhinoviruses, and Meningo viruses showing “canyons”, depression zones, or cavities in accordance with the “Canyon hypothesis” [7].

Although previous cryo-EM structural and biochemical studies on furin-cleaved and native SARS-CoV-2 S protein and bat CoV RaGT13 S protein have indicated strong similarity, the native human S protein showed higher stability and a 1000-fold higher binding affinity to the human ACE2 receptor. It suggests that furin cleavage decreased the overall S protein stability and facilitated the open conformation required for viral particle binding to the ACE2 receptor [8]. Furthermore, it has been demonstrated that the D614G mutation in the SARS-CoV-2 S protein reduced S1 shedding and increased infectivity [9]. In contrast to bat RaTG13, SARS-CoV-2 recombination presumably occurs between the S1 and S2 domains in the S protein enabling the utilization of furin protease. Despite the analysis of numerous clinical isolates of the SARS-CoV-2 S protein, no alternative recombination seems to occur, suggesting that the furin S1/S2 cleavage site is unique for recombination. Moreover, the four amino acid insertion, which creates a novel furin cleavage site, supports it. Although human Coronaviruses frequently contain “hot spots” for non-synonymous amino acid replacements affecting host tropism/adaptation, resistance to neutralizing antibodies and immune evasion [10], only a single high-frequency non-synonymous mutation (D614G) has been identified from clinical SARS-CoV-2 isolates [11]. Based on these findings, the SARS-CoV-2 S protein does not occur as a mutational “hot spot” in contrast to other human Coronaviruses.

So, the million-dollar question today is the origin of SARS-CoV-2. Why is this important? Perhaps a citation of Theodore Roosevelt is in place: “The more you know about the past the better you are prepared for the future”. Indeed, not only of scientific curiosity or finding someone to blame, but of practical awareness and preparation for potential emerging new outbreaks, the origin of SARS-CoV-2 is of utmost importance. As the current consensus within the scientific community strongly indicates, it is improbable (though not zero) that the SARS-CoV-2 emerged through laboratory manipulations. Two alternative hypotheses for the origin of SARS-CoV-2 have been presented [3]: natural selection in an animal host before zoonotic transfer or natural selection in humans following zoonotic transfer. In the earlier case, despite the high similarity between SARS-CoV-2 and bat Coronaviruses, such as bat Coronavirus RaTG13 (96% identical), there are significant discrepancies as we have pointed out. Moreover, some pangolin Coronaviruses exhibit high similarity to the RBD region in the SARS-CoV-2 S protein, including six RBD key residues [3,12]. In this latter case, the SARS-CoV-2 genome has been postulated to have adapted during undetected human-to-human transmission, which allowed the pandemic to accelerate [13]. It seems that sequence data accumulated from SARS-CoV-2 genomes so far indicate that the isolated SARS-CoV-2 genomes derived from a common ancestor. Moreover, the identification of a very similar RBD sequence in the pangolin Coronavirus S protein to the one found in SARS-CoV-2 S supports the susceptibility of transmission to humans.

In any case, additional research is required before we can confirm the origin of SARS-CoV-2. In addition to our analysis on the SARS-CoV-2 S protein, we have also targeted the SARS-CoV-2 ORF8 and ORF10 proteins in comparison to other coronaviruses to further shed some light on the origin of SARS-CoV-2, which will soon be shared with the scientific community.

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
