# Peer review of "The Importance of Research on the Origin of SARS-CoV-2"

_viruses, 2020, doi:10.3390/v12111203_

Round 1

Reviewer 1 Report

In December 2019, COVID-19 was detected, and human-to-human droplets as well as the contact transmission were reported. According to statistical reports from the World Health Organization (WHO) to October 12, 2020, there had been over 37,423,660 confirmed cases worldwide with over 1,074,817 deaths. However, there are currently no effective medications against coronaviruses, therefore to explore and understand the origin of SARS CoV-2 is of utmost importance. As mention by Theodore Roosevelt “The more you know about the past, the better you are prepared for the future”. The authors summarized two alternative hypotheses for the origin of SARS-CoV-2 had been presented, the first is natural selection in an animal host before the zoonotic transfer, since SARS-CoV-2 and bat RaTG13 showing 96% identity some pangolin Coronaviruses exhibit high similarity to particularly the receptor-binding domain of SARS-CoV-2. Whereas the other is natural selection in humans following the zoonotic transfer, that means the SARS-CoV-2 genome has adapted during human-to-human transmission, and based on available data, the isolated SARS-CoV-2 genomes derive from a common origin.  Meanwhile, the author also revealed the new targets of ORF8 and ORF10 proteins for SARS-CoV-2, which may further shed new light on the origin of SARS-CoV-2. Therefore, the current manuscript could be accepted for publication in Viruses.

Author Response

We would like to thank the reviewer for his/her comments. As there are no demands for any revision, there is nothing to add here.

Reviewer 2 Report

Brief summary

In this editorial the Authors express their own opinion on the origin of the SARS-CoV-2 virus, causing the current COVID-19 pandemic. The manuscript submitted is based on the analysis of recent scientific evidence and provides a thorough point of view on this emerging topic. The Authors conclude by emphasizing the need for further research aimed to confirm the origin of SARS-CoV-2.

Broad comments

This editorial provides an objective analysis of scientific findings aimed at revealing the proximal origin of the pandemic SARS-CoV-2. The nature of the skills possessed by the authors (bioinformatics, structural biology, molecular biology, immunology and virology) accounts for the interdisciplinary approach used to realize their recent publication “Questions concerning the proximal origin of SARS-CoV-2” (Letter in the Journal of Medical Virology, cited as reference [4] of the present editorial), in which both opinions generally accepted by most of the scientific community (naturally originated virus) and conflicting/contrary opinions (laboratory construct or intentionally engineered virus) have been discussed.

In general I recommend a few small changes which could help to improve the manuscript clarity.

Please, see “Specific comments” for details.

Specific comments:

Abstract:

  • Pag. 1_45: “..... discrepancy to other Coronaviruses exist. .....”.     I suggest to integrate this sentence by reporting the origin (please specify at least if human and/or animal) of coronaviruses showing the mentioned differences in host tropism.
  • Pag. 2_53: I suggest to integrate the sentence “..... exhibit high similarity to ......” as follows “..... exhibit very high similarity to ......”.

__________________________________________________________

  • Page 2_62: “..... as of today (September 4), ...... I suggest to add the year “September 4, 2020”.
  • Page 2_81: Please see comment to Pag. 1_45.
  • Page 2_82-83_ I think it's better to refer to specific changes “..... in host tropism .....” rather than “..... during host tropism”.
  • Page 2_85-86: To improve the clarity of the sentence For instance, the hidden glycan-binding domains located in cavities in the S protein NTD domain are not present in SARS-CoV-2, limiting their access to antibodies and immune cells [6].” I suggest to amend it as follows “For instance, the hidden glycan-binding domains located in cavities in the S protein NTD domain, limiting their access to antibodies and immune cells, are not present in SARS-CoV-2 [6]”.
  • Page 2_90-94 “Although previous cryo-EM structural and biochemical studies on furin-cleaved and native SARS-CoV-2 S protein and bat CoV RaGT13 S protein have indicated strong similarity, the native human S protein showed higher stability and a 1000-fold higher binding affinity to the human ACE2 receptor [8]. It suggests that furin cleavage decreased the overall S protein stability and facilitated the open conformation required for viral particle binding to the ACE2 receptor”. I suggest to move the reference citation [8] from line 93 to line 94, at the end of the second sentence.
  • Page 3_117: I suggest to cite both related references [3,12] in this line. These two papers refer to six and five RBD key residues, respectively.

Author Response

We would like to thank the reviewer for the valuable comments and we have addressed all the minor points addressed following the suggestions made by the reviewer.